# Cyclin-Dependent Kinase 18 Controls Trafficking of Aquaporin-2 and Its Abundance through Ubiquitin Ligase STUB1, Which Functions as an AKAP

**DOI:** 10.3390/cells9030673

**Published:** 2020-03-10

**Authors:** Alessandro Dema, Dörte Faust, Katina Lazarow, Marc Wippich, Martin Neuenschwander, Kerstin Zühlke, Andrea Geelhaar, Tamara Pallien, Eileen Hallscheidt, Jenny Eichhorst, Burkhard Wiesner, Hana Černecká, Oliver Popp, Philipp Mertins, Gunnar Dittmar, Jens Peter von Kries, Enno Klussmann

**Affiliations:** 1Max Delbrück Center for Molecular Medicine in the Helmholtz Association (MDC), Robert-Rössle-Strasse 10, 13125 Berlin, Germany; alessandro.dema.88@gmail.com (A.D.); DoerteFaust@gmx.net (D.F.); kerstin.zuehlke@mdc-berlin.de (K.Z.); andrea.geelhaar@mdc-berlin.de (A.G.); tamara.pallien@mdc-berlin.de (T.P.); eileen.hallscheidt@me.com (E.H.); cernecka.h@gmail.com (H.Č.); Oliver.Popp@mdc-berlin.de (O.P.); Philipp.Mertins@mdc-berlin.de (P.M.); gunnar.dittmar@lih.lu (G.D.); 2Leibniz-Forschungsinstitut für Molekulare Pharmakologie (FMP), Robert-Rössle-Strasse 10, 13125 Berlin, Germany; Lazarow@fmp-berlin.de (K.L.); M.Wippich@web.de (M.W.); Neuenschwander@fmp-berlin.de (M.N.); eichhorst@fmp-berlin.de (J.E.); burkhard.wiesner@gmail.com (B.W.); kries@fmp-berlin.de (J.P.v.K.); 3DZHK (German Centre for Cardiovascular Research), Partner Site Berlin, Oudenarder Strasse 16, 13347 Berlin, Germany

**Keywords:** CDK18, AQP2, PKA, AKAP, STUB1, CHIP

## Abstract

Arginine-vasopressin (AVP) facilitates water reabsorption in renal collecting duct principal cells through regulation of the water channel aquaporin-2 (AQP2). The hormone binds to vasopressin V2 receptors (V2R) on the surface of the cells and stimulates cAMP synthesis. The cAMP activates protein kinase A (PKA), which initiates signaling that causes an accumulation of AQP2 in the plasma membrane of the cells facilitating water reabsorption from primary urine and fine-tuning of body water homeostasis. AVP-mediated PKA activation also causes an increase in the AQP2 protein abundance through a mechanism that involves dephosphorylation of AQP2 at serine 261 and a decrease in its poly-ubiquitination. However, the signaling downstream of PKA that controls the localization and abundance of AQP2 is incompletely understood. We carried out an siRNA screen targeting 719 kinase-related genes, representing the majority of the kinases of the human genome and analyzed the effect of the knockdown on AQP2 by high-content imaging and biochemical approaches. The screening identified 13 hits whose knockdown inhibited the AQP2 accumulation in the plasma membrane. Amongst the candidates was the so far hardly characterized cyclin-dependent kinase 18 (CDK18). Our further analysis revealed a hitherto unrecognized signalosome comprising CDK18, an E3 ubiquitin ligase, STUB1 (CHIP), PKA and AQP2 that controls the localization and abundance of AQP2. CDK18 controls AQP2 through phosphorylation at serine 261 and STUB1-mediated ubiquitination. STUB1 functions as an A-kinase anchoring protein (AKAP) tethering PKA to the protein complex and bridging AQP2 and CDK18. The modulation of the protein complex may lead to novel concepts for the treatment of disorders which are caused or are associated with dysregulated AQP2 and for which a satisfactory treatment is not available, e.g., hyponatremia, liver cirrhosis, diabetes insipidus, ADPKD or heart failure.

## 1. Introduction

Arginine-vasopressin (AVP) activates vasopressin V2 receptors (V2R) on the surface of renal collecting duct principal cells and triggers the redistribution of the water channel aquaporin-2 (AQP2) from intracellular vesicles into the plasma membrane. The membrane insertion facilitates water reabsorption from primary urine and fine-tunes body water homeostasis [1,2,3]. Dysregulation of the system has important clinical implications. For example, mutations in the V2R or the AQP2 gene cause diabetes insipidus, a disease characterized by an excessive excretion of hypotonic urine (up to 20 l/day); elevated levels of AVP can cause or are associated with e.g., hyponatremia, liver cirrhosis or heart failure. In autosomal dominant polycystic kidney disease (ADPKD), high cAMP levels in the renal principal cells contribute to the excessive water retention through a predominant localization of AQP2 in the plasma membrane. Satisfactory treatments of such disorders are not available [4,5]. Thus, elucidation of the molecular mechanisms underlying the regulation of AQP2 will not only contribute to understanding the molecular mechanisms underlying this exocytosis-like process but may also pave the way to new concepts for the treatment of the above-mentioned water balance disorders.

AVP binding to V2R stimulates synthesis of cAMP, which activates a perinuclear pool of protein kinase A (PKA) where a major portion of AQP2-bearing vesicles resides under resting conditions [6]. PKA is tethered to AQP2-bearing vesicles by the A-kinase anchoring proteins (AKAPs) AKAP18δ and AKAP220 [6,7,8,9]. The anchoring of PKA by AKAPs is required for the AVP-induced redistribution of the AQP2 into the plasma membrane [10]. The AVP-mediated activation of PKA was long thought to cause phosphorylation of AQP2 at serine (S) 256 and thereby to trigger the trafficking of the AQP2-bearing vesicles to the plasma membrane [11,12,13,14,15]. However, S256 can be phosphorylated in the absence of PKA [16] and by other kinases, including calmodulin-regulated kinase 2 (CAMK2), serum/glucocorticoid-regulated kinase (SGK) and protein kinase G (PKG) [17,18]. It is now clear that S256 phosphorylation interferes with the endocytic retrieval of AQP2 from the plasma membrane [16,19,20,21]. The AVP stimulation of the V2R is followed by further changes of the phosphorylation of AQP2, for example at S264 and S269. The former may be involved in the control of AQP2 trafficking through the endosomal system, [11] the latter extends the localization of AQP2 in the plasma membrane, most likely because it decreases the rate of endocytosis [12,14,22,23,24]. The kinases phosphorylating S264 and S269 are not known. Under resting conditions, AQP2 is phosphorylated at S261, which can amongst other kinases be catalyzed by p38-mitogen-activated protein kinase (p38-MAPK). The S261 phosphorylation is associated with AQP2 ubiquitination and proteasomal degradation. The AVP-mediated activation of PKA causes inhibition of p38-MAPK and a reduction of the S261 phosphorylation and ubiquitination. The result is an increase in the protein abundance and plasma membrane localization of AQP2 [5,25]. Thus, an AVP/PKA-induced network of kinases both triggers the redistribution of AQP2 into the plasma membrane and controls its abundance. However, the signaling downstream of the AVP-mediated PKA activation is unclear.

We sought to identify kinases that control the localization of AQP2 by a cell-based siRNA screen targeting 719 kinase-related genes that represent the majority of the kinases of the human genome. This approach identified a hitherto unrecognized signalosome comprising the so far hardly characterized cyclin-dependent kinase 18 (CDK18), an E3 ubiquitin ligase, STUB1 (CHIP), PKA and AQP2 that controls both the localization and abundance of AQP2.

## 2. Materials and Methods

### 2.1. Kinome Screening and Automated Immunofluorescence Microscopy

The mouse protein kinases siRNA sub-library is part of the Mouse siGENOME siRNA library-SMART pool (Horizon inspired cell solutions/Dharmacon, https://dharmacon.horizondiscovery.com, 0.25 nmol). The kinase sub-library targets 719 kinases out of the 16,872 genes targeted by the genome-wide library. Each gene expression is silenced with a pool of four different siRNAs (19 nucleotides) directed against different positions of one mRNA. The kinase sub-library was distributed over three 384-well microtiter plates. As controls, non-targeting siRNA (siNT) and cytotoxicity-inducing TOX siRNA were included.

Mouse collecting duct (MCD)4 cells, [5,26,27] stably expressing human AQP2, were seeded in 384-well ultrathin optical bottom (190 µm) microtiter plates (GREINER, µclear). The MCD4 cells were transfected with the siRNAs using Lipofectamine 2000 (Thermo Fisher; https://www.thermofisher.com). Three days post transfection, the cells were left untreated or stimulated with forskolin.

As previously described, AQP2 was visualised with primary antibody H27 and Cy3-conjugated mouse anti-rabbit IgG (Jackson ImmunoResearch Laboratories, #211-165-109) [28]. Nuclei were stained with DAPI. Microscopic analysis was conducted within seven days after staining to avoid time-dependent signal decrease [29]. Images were acquired using an ArrayScan V^TI^ HCS Reader automated microscope (Thermo Fisher Scientific/Cellomics, Bonn, DE).

Candidates whose knockdown prevented the forskolin-induced redistribution of AQP2 where identified by image analysis using CellProfiler (https://cellprofiler.org) [30] and KNIME (https://www.knime.com) software. Cellular phenotypes were analyzed via conventional z-score analysis and classified using a KNIME-integrated multilayer perceptron (MLP). MLP is a powerful machine learning model that autonomously identifies complex data patterns and maps them to defined groups. It belongs to the class of artificial neuronal networks that can be used for gene identification approaches [31]. Machine learning algorithms can be trained for recognizing phenotypes [32]. MLP was trained with CellProfiler-derived metadata of unstimulated and forskolin-stimulated mock-transfected cells, showing many or few perinuclear AQP2 speckles, respectively. Network quality was validated on data from unstimulated and FSK-stimulated siRNA-transfected controls. Data of each microtiter plate were classified separately (Appendix A). The metadata of one microtiter plate were also analyzed by z-score and hits relative to other siRNA knockdowns were identified. z-factors of the remaining plates were <0.5 and hence, z-score analysis not feasible. A siRNA pool was regarded inhibitory if it scored >2 standard deviations above the mean, i.e., if the z-score was larger than 2. No microtiter plate- or well-specific effects were observed and thus systemic errors could be excluded.

### 2.2. Cell Culture, Transfection and Generation of Expression Plasmids

MCD4 cells were cultured in 5% FCS, 5 µM Dexamethasone F12-DMEM GlutaMAX (Thermo Fisher). For siRNA and DNA transfection, Lipofectamine 2000 and Lipofectamine 3000 (both Thermo Fisher), respectively, were used according to the supplier’s protocol. HEK293T were cultured in 10% FCS, 1% PS DMEM GlutaMAX (Thermo Fisher) and transfected using Lipofectamine 2000.

Primary mouse inner medullary collecting duct (mIMCD) cells were derived and cultured according to our protocol for rat IMCD cells [33] and transfected using Lipofectamine 3000 according to the manufacturer’s instructions.

CDK18 siRNA (#M-040145-01-0005/18557)), STUB1 (#M-007201-02-0005) and non-targeting siRNA (siNT; NT#2) (against firefly luciferase; #D-001206-14-20) were purchased as siGENOME SMART pools (Thermo Fisher).

The following expression plasmids were generated on the basis of vector pCMV6ev (Origene, Rockville, MD, USA): pCMV6-CDK18-FLAG-Wt, pCMV6-CDK18-FLAG-K150R (kinase dead) and pCMV6-CDK18-FLAG-S12D (constitutively active). The plasmid pCIneo-3HA-mCycA2 was kindly provided by Dr. Wiebusch and Virchow-Klinikum, Berlin. The cDNA encoding CDK18 Wt (gene art string; Thermo Fisher) was inserted into pCMV6ev (Origene) using the restriction sites HindIII (5′) and XhoI (3′). In order to obtain the S12D and K150R mutants, mutagenesis was carried using the Quick Change II system (Agilent Technologies, Santa Clara, CA, USA) and the primers: 

S12D F: GGGGCACGGACAGGTCAAGCCTTCGCTTGAAG; S12D R: CTTCAAGCGAAGGCTTGACCTGTCCGTGCCCC; K150R F: AGAACCTCGTGGCCCTGAGGGAGATCCGG; K150R R: CCGGATCTCCCTCAGGGCCACGAGGTTCT

### 2.3. Laser Scanning Microscopy

MCD4 or primary mIMCD cells were treated with forskolin (30 µM, 30 min), the heat-stable and stearate-coupled PKA inhibitor PKI (10 µM, 30 min), MG132 (10 µM, 3 h) or combinations thereof. Immunofluorescence microscopy was performed with antiserum H27 against AQP2 as described.^25^ Nuclei were stained with DAPI (Hoffmann-La Roche) and fluorescence signals were visualized using a LSM 780 microscope (Zeiss, Oberkochen, Germany).

### 2.4. Cell Lysis, Immunoprecipitation, cAMP Pull-Down, Western Blotting and RII Overlay

For Western blot analysis, cells were lysed using standard lysis buffer (SLB, 10 mM K_2_HPO_4_, 150 mM NaCl, 5 mM EDTA, 5 mM EGTA, 1% Triton X-100, 0.2% Na-deoxycholate, pH 7.4) [34] supplemented with protease and phosphatase inhibitors (Roche, https://www.roche.de). Where indicated, cell lysates were incubated with peptide Ht31, the inactive control PP-Ht31 (each 30 µM, 30 min, 30 °C); dbcAMP was added to freshly isolated murine inner medullary homogenates (1 mM, 30 min, 37 °C). 

For immunoprecipitations, cells were lysed in mild lysis buffer (MLB, 137 mM NaCl, 10 mM Na_2_HPO_4_, 2.7 mM KCl, 1.8 mM KH_2_PO_4_, 0.2% Triton X-100, 2 mM EDTA, 2mM EGTA, pH 7.4) supplemented with protease and phosphatase inhibitors, and 500 µg protein aliquots were incubated for 1–3 h at 4 °C with 50 ul of Protein A-conjugated magnetic Dynabeads (Thermo Fisher), loaded with the appropriate antibody. After 3 washes in MLB, the precipitated immunocomplexes were eluted from the beads with of 0.1 M glycine, pH 2.4, neutralized with 1M Tris-HCl, pH 10.6, diluted in Laemmli sample buffer and analyzed by Western blotting. For immunoprecipitations, the following antibodies were used: AQP2 H27 (rabbit serum, custom-made), [35] STUB1 (rabbit, ab134064; Abcam, Cambridge, UK), CDK18/PCTAIRE3 (rabbit, C-17; Santa Cruz Biotechnology, Heidelberg, Germany).

For cAMP-agarose precipitations, lysates were incubated with 8-AHA-cAMP agarose (4 °C, 3 h; Biolog, Bremen, Germany); agarose-bound proteins were washed four times with MLB and eluted with Laemmli sample buffer. cAMP (15 mM) was used to abolish the binding of PKA to the beads [36].

Western Blotting was carried out as described [7,34,36] with the following antibodies: HSP90 (mouse, AC88; Enzo Life Sciences, Lörrach, Germany), pan-Cadherin (mouse; Sigma-Aldrich, Saint Louis, USA), GAPDH (rabbit, #2118) STUB1/CHIP (rabbit, #2080), p38 MAPK (rabbit, #2371), phospho p38 MAPK (T180/Y182; rabbit, #4511), Ubiquitin (rabbit, #3936), phospho-PKA substrate (rabbit, #9624; all Cell Signaling Technology, Frankfurt/Main, Germany), PKA regulatory RIIα and catalytic subunits (all mouse; BD, Heidelberg, Germany), CDK18/PCTAIRE3 (rabbit), AQP2 (goat, C-17; Santa Cruz Biotechnology), STUB1/CHIP (mouse, ab180038), APQ2 pS261 (rabbit, ab72383; Abcam), phosphoS256 AQP2 (rabbit, ZDE12147_0798; Eurogentec, Köln, Germany), HA high affinity (rat, 1867423; Hoffmann-La Roche, Basel, Switzerland). Signals were detected using EMD Millipore Immobilon Western Chemiluminescent HRP Substrate (Thermo Fisher) and an Odyssey Fc Imaging System (LI-COR Biotechnology, Lincoln, USA). The semi-quantitative densitometric evaluation was performed with the Image Studio Lite software (LI-COR Biosciences, Bad Hombug, Germany). For RII overlay [10] RIIα was purified and labelled with ^32^P as described. [10,37]

### 2.5. Kinase and pNPP Phosphatase Activity Assays

PKA activity was monitored using the PepTag Assay (Promega, Walldorf, Germany) as described. [5,26] For non-radioactive evaluation of CDK18 kinase activity, Wt, constitutively active and kinase dead forms (see above) and cyclin A were transiently expressed in HEK293 cells. The CDK18 forms were immunoprecipitated 48 h later with anti-FLAG-coupled magnetic beads (Sigma-Aldrich; 1 h, 4 °C). The beads were washed and incubated with individual spot-synthesized peptides (see below; 1 h, 30 °C) in Kinase Assay Buffer (50 mM Tris-HCl, pH 7.5, 20 mM MgAc, 50 µM ATP). The peptide spots were washed in TBS-Tween, incubated with primary anti-phospho-Ser-CDKs substrate antibody (rabbit #2324 CST). Phosphorylation was visualized by Western blotting.

For pNPP-based phosphatase assays, cells were lysed in MLB without phosphatase inhibitors, and 100 µg of protein were diluted in 10 mM pNPP Colorimetric Assay Buffer as described. [5,38] After 30 min incubation at 30 °C, the absorbance at 405 nm was determined with an xMark™ Microplate Absorbance Spectrophotometer (Bio-Rad Laboratories, Feldkirchen, Germany), and phosphatase activity was then calculated. 

### 2.6. Synthetic and Spot-Synthesised Peptides

Quantities of 20- or 25-mer peptides were spot-synthesized as described [34,36,39,40] using an Intavis ResPep SL spot synthesizer (Intavis, Cologne, Germany). The spot-synthesized sequence of amino acids 792–818 of RB: FPS**S**PLRIPGGNIYI**S**PLK**S**PYKISEG. All three highlighted S within the SP motifs are potential substrates for CDK18.

### 2.7. Sample Preparation for Mass Spectrometry

For mass spectrometric analysis, samples were run on a stacking SDS-PAGE. After Coomassie Brilliant Blue staining (G250), gel pieces were minced to 2 mm^3^ pieces and digested with trypsin, based on Shevchenko et al. [41] in an automated fashion using a PAL robot (CTC Analytics/Axel Semrau, Sprockhövel, Switzerland). Briefly, gel pieces were washed with 50% ethanol in 50 mM ammonium bicarbonate (ABC) and 50 mM ABC in an alternating fashion. Reduction of disulfide bonds was carried out using 2.5 pmol tris (2-carboxyethyl) phosphine (TCEP) and alkylation using 12.5 pmol chloroacetamide both for 30 min at room temperature followed by a 10 h digest with 5 µg sequencing grade trypsin (Promega). Peptides were extracted with extraction buffer (80% acetonitrile, 0.1% (v/v) formic acid) and subsequently dried in a speed-vac followed by purification on C_18_ stage-tips [42]. The eluted peptides were dried in a speed-vac and resuspended in 3% acetonitrile, 0.1% (v/v) formic acid for LC–MS measurement.

Samples were measured by LC–MS/MS on a Q Exactive Plus mass spectrometer (Thermo Fisher) connected to an EASY n-LC system (Thermo Fisher). A volume of 5 μL sample was injected on a trap-free system and a 45 min gradient (4% to 76% acetonitrile) was applied. The peptides were separated on an in-house prepared nano-LC column (0.074 × 250 mm, 3 µm Reprosil C_18_, Dr Maisch GmbH, Ammerbuch-Entringen, Germany) using a flow rate of 0.25 μL/min. MS acquisition was performed with an MS1 resolution of 70,000 and a scan range from 300 to 1700 m/z. The top 10 peaks were selected for fragmentation in data-dependent mode using an MS2 resolution of 17,500 and a maximum injection time of 60 ms. Dynamic exclusion was set to 20 s and the normaliezd collision energy was specified to 26.

For analysis, the MaxQuant software package version 1.5.2.8 [43] was used. Carbamidomethylation on cysteine was set as a fixed modification and oxidized methionine, acetylated N-termini and phosphorylation as variable modifications. An FDR of 0.01 was applied for peptides and proteins and the Andromeda search [44] was performed using a mouse Uniprot database (July 2018).

MS intensities were normalized by the MaxLFQ algorithm implemented in MaxQuant [45] while using the match-between-runs feature. Further data analysis was done using R and the Protigy package (https://github.com/broadinstitute/protigy). Log_2_-transformed MaxLFQ-normalized intensities among the replicates of the groups to be related were used for comparison applying a moderated t-test. [46] After removal of potential contaminants and proteins that have been only identified by a modified peptide from the protein list, a valid-value filter was applied, requiring at least three valid values in one of the groups to be compared. The resulting list was used for imputation using a row-wise Gaussian distribution with a width of 0.2 and a downshift of 1.8. Proteins were considered significantly enriched with a Benjamini–Hochberg-adjusted *p*-value smaller than 0.05 (i.e., 5% FDR).

### 2.8. Statistical Analysis

Statistical analysis was performed using GraphPad Prism5 and 7 (GraphPad, La Jolla, CA, USA). Unpaired t-test or one-way ANOVA with post hoc Bonferonni were applied. Significant differences are indicated as * *p* ≤ 0.05, ** *p* ≤ 0.01 or *** *p* ≤ 0.001 or **** *p* ≤ 0.0001. Mean plus standard error of mean (SEM) are plotted.

## 3. Results

### 3.1. Kinome Knockdown Identifies Candidates Controlling the Redistribution of AQP2 to the Plasma Membrane

We sought to identify proteins downstream of PKA that control AQP2 trafficking by siRNA-mediated silencing of the kinome (719 genes) of Murine Collecting Duct 4 (MCD4) cells (Figure 1A, Appendix A). They represent a cellular model for AQP2 trafficking. The cells stably express human AQP2 and when stimulated with the adenylyl cyclase activator, forskolin, cAMP levels increase, and trigger the AQP2 redistribution predominantly to the basolateral plasma membrane [5,26,27]. The AQP2 redistribution was monitored by automated immunofluorescence microscopy. We used image analysis software CellProfiler [30] and KNIME (knime.org) to identify candidates whose knockdown prevented the redistribution of AQP2. Our approach identified 19 hits (Table 1; Appendix A). The knockdown of 6 candidates decreased cell viability to less than 60% compared to cells transfected with control, non-targeting siRNA (siNT) (Table 1, Appendix A). Since the expression of those genes was necessary for MCD4 cell survival, these candidates were excluded from further analysis. The remaining 13 candidates reduced MCD4 cell viability to 64%–97% (Table 1). None of them was previously related to the control of AQP2. Some of the candidates may have indirect links to signaling proteins and pathways known to be involved in AQP2 regulation. For example, PKIA blocks PKA activity by binding to its catalytic subunits upon their cAMP-dependent dissociation from regulatory subunits [47]. It regulates the nuclear export of the free catalytic PKA subunits [48]. STK11 controls the activity of AMP-activated protein kinases (AMPK) and thus, is involved in various processes such as cell growth, energy metabolism and cell polarity [49]. An involvement of PKA and AMPA in AQP2 regulation was already known.

### 3.2. Knockdown of CDK18 Inhibits the cAMP-Induced Redistribution of AQP2

Among the 13 candidates was cyclin-dependent kinase (CDK)18, one of the highly homologous PCTAIRE kinases, comprising PCTAIRE1/CDK16, PCTAIRE2/CDK17 and PCTAIRE3/CDK18. The function of CDK18 is largely unknown. The inhibitory effect of its knockdown on the accumulation of AQP2 at the plasma membrane suggested that it may be involved in a new pathway controlling AQP2. Therefore, we aimed to characterize its function in the regulation of AQP2. It has been shown that CDK18 may be involved in Alzheimer’s disease, [50,51] in the Ataxia Telangiectasia and RAD3-related (ATR)-mediated response to single strand DNA and in cell cycle control [52,53]. Indeed, its role in cell cycle control is in line with the by 30% reduced MCD4 cell number if CDK18 was knocked down (Table 1).

The knockdown of CDK18 to around 10% of its expression level with either a pool of four or with a single siRNA inhibited the forskolin-induced redistribution of AQP2 to the plasma membrane (Figure 1B,C) while it hardly affected AQP2 localization in resting MCD4 cells (Figure 1B). Thus, CDK18 plays a role not only in controlling viability/proliferation, but also in the control of the cAMP-dependent trafficking of AQP2. A STRING analysis did not indicate a link of CDK18 with the other identified hits (data not shown).

### 3.3. AVP Induces PKA-Dependent Increases of CDK18 Phosphorylation and Abundance 

PKA phosphorylates CDK18 at S12, S66 and S109. The phosphorylation at S12 increases CDK18 activity even in the absence of the activating cofactor Cyclin A2 [54]. We evaluated the PKA phosphorylation of CDK18 by immunoprecipitation of the kinase and detection by Western blotting with phospho-PKA substrate antibodies. The antibody recognizes the consensus motif RRXS*/T* in PKA substrates when the serine or the threonine is phosphorylated. All CDK18 PKA sites can be recognised by the antibody. Forskolin stimulation of MCD4 cells increased CDK18 phosphorylation. The increase was prevented with a cell-permeant (stearate-coupled) heat-stable, specific PKA inhibitor (S-PKI), confirming that PKA is the responsible kinase (Figure 2A). In a physiological system, in murine renal inner medullae, AVP increased the phosphorylation of CDK18; again, this was prevented by S-PKI and thus catalysed by PKA (Figure 2B). In addition, AVP increased the protein abundance of CDK18, which was also prevented by S-PKI. The data indicated that the cAMP-mediated activation of PKA increased CDK18 kinase phosphorylation and its protein abundance, and presumably CDK18 activity (Figure 2B).

### 3.4. CDK18 Controls the Phosphorylation of AQP2 at S261, Its Ubiquitination and Abundance

Next, we investigated how the cAMP/PKA-dependent modulation of CDK18 affected AQP2. In MCD4 cells, the knockdown of CDK18 did not affect the AQP2 mRNA level or the phosphorylation of S256. The cAMP elevation through forskolin did also not affect the S256 phosphorylation, which is in line with the observation that the plasma membrane accumulation of AQP2 can occur in the absence of PKA [16]. However, upon CDK18 knockdown the S261 phosphorylation of AQP2 decreased and its protein abundance increased both in the absence or presence of forskolin (Figure 2C).

To investigate whether CDK18 phosphorylates AQP2 at S261, FLAG-tagged CDK18 as wild type (Wt), constitutively active (Ser12Asp; S12D, mimicking activation by PKA-mediated phosphorylation) and kinase dead (Lys150Arg; K150R) forms [54] were expressed in and purified from HEK293 cells through precipitation via their Flag tags. The activity of the Wt CDK18 and the constitutively active S12D versions were confirmed by their ability to phosphorylate a spot-synthesised peptide substrate derived from the retinoblastoma (Rb) protein (Figure 3A). The precipitated constitutively active CDK18 also phosphorylated S261 of AQP2 immunoprecipitated from resting MCD4 cells (Figure 3B).

We had previously observed that a decrease in the phosphorylation of AQP2 at S261 is associated with a decrease in its poly-ubiquitination and an increase in its abundance [5,25]. The observation that the knockdown of CDK18 decreased the S261 phosphorylation (Figure 2C) and that CDK18 phosphorylated S261 (Figure 3B) suggested that CDK18 also influences the ubiquitination of AQP2. Indeed, the CDK18 knockdown was associated with a decrease in the poly-ubiquitination of AQP2 (Figure 4A), explaining the increase in its abundance (Figure 2C), and indicating that CDK18 controls a downstream E3 ubiquitin ligase.

### 3.5. CDK18 Controls AQP2 Ubiquitination and Abundance through the E3 Ubiquitin Ligase STUB1 

In order to identify the E3 ligase that is regulated by CDK18 to ubiquitinate AQP2, we compared the interactomes of the kinase dead and constitutively active CDK18 forms (see Figure 3A and Appendix A) by mass spectrometry (Figure 4B). The constitutively active form of CDK18 was associated with the E3 ubiquitin ligase STUB1 (STIP1 homology and U-box containing protein 1) also termed CHIP (C-terminal Hsc70 interacting protein) [55] (Figure 4B). CHIP1 has previously been identified as an E3 ubiquitin ligase of AQP2 [56,57]. Therefore, and as expected, STUB1 knockdown in MCD4 cells decreased the poly-ubiquitination of AQP2 and increased its abundance (Figure 4C). The knockdown also mimicked the inhibitory effect of the CDK18 knockdown on the AQP2 redistribution to the plasma membrane (Figure 4D). Surprisingly, the knockdown of CDK18 decreased the protein abundance of STUB1 (Figure 4E). This was independent of transcription, since the mRNA level was not affected by the knockdown (Figure 4F), and it was not catalyzed by the proteasome as the decrease in STUB1 protein abundance was not abrogated by the proteasome inhibitor, MG132 (Figure 4G). In primary murine inner medullary collecting duct principal cells, the knockdown of CDK18 also reduced STUB1 protein levels (Figure 4H), demonstrating the physiological relevance of the mechanism. The STUB1 knockdown did not affect AQP2 phosphorylation at S261 and S256 or the CDK18 abundance (Appendix A).

### 3.6. A CDK18-Containing Signalosome with AQP2 and PKA Is Organised by STUB1, Which Functions as an AKAP

The STUB1 knockdown caused a decrease of the interaction of AQP2 and CDK18 in MCD4 cells (Figure 4C), suggesting that the AQP2 regulation involves a larger protein complex including AQP2/CDK18 and STUB1 (Figure 4B,C). The other observation, that PKA phosphorylates CDK18 (Figure 2), prompted us to investigate whether PKA is also associated with the protein complex. PKA holoenzyme consists of two regulatory R subunits (RIα, RIβ, RIIα or RIIβ) each of which binds one catalytic C subunit. cAMP binding to the R subunits induces dissociation of the C subunits, which then phosphorylate their substrates [58]. AKAPs directly interact with regulatory subunits of PKA and tether the AKAP-PKA complex to a defined cellular compartment. The interaction between PKA and AKAPs is mediated by a conserved amphipathic α-helix of AKAPs, the A-Kinase Binding (AKB) domain [59,60,61,62]. Peptides such as Ht31 or AKAP18δ (L314E), derived from the AKB domains of AKAP-Lbc and AKAP18δ, respectively, bind regulatory PKA subunits with sub-nanomolar affinity and thereby block the interaction with AKAPs [63,64]. cAMP-agarose can be used to precipitate R subunits of PKA by virtue of their ability to bind cAMP, and proteins bound to the PKA subunits. To examine whether PKA interacts with the AQP2/CDK18/STUB1 complex and whether this would occur in an AKAP-like manner through an AKB domain, we precipitated R subunits of PKA with cAMP-agarose from MCD4 cells (Figure 5A). As expected, the precipitates contained RII (RIIα) subunits which were undetectable if the samples were incubated with an excess of cAMP. The cAMP displaces the R subunits from the beads. The cAMP-agarose precipitates also contained AQP2. The amount of AQP2 in the precipitates was reduced if the samples were treated with the peptide Ht31, indicating that an AKAP mediates the interaction between PKA and AQP2 (Figure 5A). When STUB1 was knocked down in MCD4 cells and cAMP-agarose pull-downs were performed, a decrease of the AQP2 in the precipitate was revealed, suggesting that STUB1 bridges the interaction between PKA and AQP2 (Figure 5B) and that it functions as an AKAP. The option that STUB1 is the AKAP tethering PKA to the AQP2/CDK18/STUB1/PKA complex was further explored by a combination of peptide spot synthesis and RII overlay. The entire amino acid sequence of STUB1 was spot-synthesised as 20-mer overlapping peptides. Recombinant radioactively labelled RIIα subunits of PKA bound to amino acids 150–190 of STUB1, indicating a direct interaction. The binding was reduced by peptide, L314E (Figure 5D). Moreover, substitution of individual amino acids in the binding region with proline, which disrupts the amphipathic helix of the AKB, [64,65] also prevented the interaction with RIIα subunits (Appendix A). Thus, STUB1 functions as an AKAP tethering PKA to the AQP2/CDK18/STUB1/PKA complex.

Finally, we tested whether the STUB1-bound PKA is involved in the cAMP-dependent regulation of AQP2. STUB1 co-immunoprecipitated with CDK18, AQP2 and regulatory RIIα subunits of PKA from resting MCD4 cells (Figure 5E). The catalytic subunits co-immunoprecipitated to a significantly lesser extent from forskolin-treated cells than from the control cells, indicating that STUB1-associated PKA within the AQP2/CDK18/STUB1/PKA multiprotein complex is activated upon cAMP elevation in MCD4 cells, i.e., by the main regulator of AQP2 (Figure 5B).

### 3.7. A Cytosolic Pool of PKA Phosphorylates CDK18 - PKA within the AQP2/CDK18/STUB1/PKA Complex Does Not Coordinate PKA Phosphorylation of CDK18 or AQP2

Next, we addressed the function of the STUB1-associated PKA. We examined whether STUB1-associated PKA mediates the phosphorylation of CDK18. For this, STUB1 expression was knocked down in MCD4 cells and the cells were stimulated with forskolin. Forskolin induced the phosphorylation of CDK18 as detected in CDK18 immunoprecipitates with phospho-PKA substrate antibody. However, the increase was not different from the increase that occurred when the cells were treated with non-targeting siRNA as a control (Figure 6A).

Moreover, treatment of MCD4 cells with a stearate-coupled and thus membrane-permeant version of the peptide Ht31, did not affect the forskolin-induced phosphorylation of CDK18 that was detected in CDK18 immunoprecipitates with our phospho-PKA substrate antibody when compared to cells that were not treated with Ht31. The peptide did, however, inhibit the PKA phosphorylation of GSK3β which is facilitated by PKA bound to the AKAP GSKIP [36] (Figure 6B). Therefore, the phosphorylation of CDK18 is catalyzed by a soluble, non-AKAP-bound pool of PKA.

We had previously shown that ubiquitin-mediated degradation of PKA subunits is a way to control PKA activity [66]. In addition, STUB1 can mediate the ubiquitination and thus degradation of catalytic PKA subunits [67,68]. However, in MCD4 cells the knockdown of STUB1 did not alter PKA subunit abundance (Figure 5C). Thus, PKA within the AQP2/CDK18/STUB1/PKA complex did not seem to control the components of the complex.

### 3.8. CDK18 Is Involved in Feedback Control of PKA and Phosphatase Activity

Our data so far indicated that forskolin stimulation of MCD4 cells caused a PKA-dependent phosphorylation of CDK18 and that the responsible pool of PKA is cytosolic and independent of the AQP2/CDK18/STUB1/PKA complex. We further enquired the functional link between CDK18 and PKA, and tested whether vice versa CDK18 is involved in the control of PKA. For this, we knocked down CDK18 with our siRNA in MCD4 cells and measured PKA activity using the Peptag assay. Forskolin induced an increase in PKA activity in untransfected MCD4 cells and in cells transfected with non-targeting or CDK18 siRNA. Of note, in MCD4 cells where CDK18 was knocked down, PKA activity was increased to a higher level than in untransfected cells or cells transfected with NT siRNA (Figure 7A). Therefore, CDK18 is involved in a negative feedback loop involving PKA and CDK18.

Our mass spectrometric analysis revealed an increased interaction of protein phosphatases with the constitutively active CDK18 (Figure 4B). This raised the possibility that CDK18 also affects protein phosphatase activity. Indeed, the knockdown of CDK18 caused a decrease of the total phosphatase activity in forskolin-treated MCD4 cells. Under resting conditions, the knockdown did not influence phosphatase activity (Figure 7B).

## 4. Discussion

Our kinome screening identified 13 hits whose knockdown inhibited the accumulation of AQP2 in the plasma membrane. We focused our analysis on the role of the so far hardly characterized CDK18 and revealed an unusual role of a cyclin-dependent kinase. CDK18 controls the redistribution of AQP2 from intracellular domains to the plasma membrane, and in concert with the E3 ubiquitin ligase, STUB1, is involved in controlling AQP2 abundance (Figure 8). A tight control of both the AQP2 localisation and abundance contributes to the fine-tuning of body water homeostasis. 

We demonstrate that S261 of AQP2 is a target for CDK18 phosphorylation. The knockdown of CDK18 reduces S261 phosphorylation. The phosphorylation of this site is associated with the ubiquitination of AQP2, which, in turn, controls the abundance of this water channel through proteasomal degradation [5,25]. Consistently, the CDK18 knockdown decreases ubiquitination and increases the abundance of AQP2. Our data indicated that CDK18 regulates STUB1 protein abundance and through this coordinates the ubiquitination of AQP2. In line with findings by Wu et al., our STUB1 (CHIP) knockdown reduced, but did not abolish, AQP2 ubiquitination [56]. Additionally, overexpression of STUB1 does not lead to a complete degradation of AQP2 [57]. Therefore, other E3 ubiquitin ligases are involved. One could be TRIM27, which was amongst the hits in our screening. Bioinformatics identified at least 10 further E3 ligases that are potentially involved in AQP2 ubiquitination [56,69]. Moreover, the knockdown of CDK18 did not abolish S261 phosphorylation, which is in line with previous findings that other kinases such as p38MAPK or ERK are involved in the phosphorylation of this site [25,70]. The residual S261 phosphorylation likely contributes to the ubiquitination observed if STUB1 expression was knocked down.

The ubiquitination of AQP2 is also involved in controlling the localization of AQP2. It is associated with AQP2 removal from the plasma membrane, followed by recycling, proteasomal degradation or excretion [1,25,71,72]. In MDCK cells, AQP2 deficient in ubiquitin binding is mainly located in the plasma membrane [72]. Therefore, knockdown or inhibition of CDK18 or STUB1, which both decreases the ubiquitination, would not be expected to interfere with the redistribution of AQP2. However, the knockdown of either CDK18 or STUB1 inhibits the cAMP-induced accumulation of AQP2 in the plasma membrane, showing that both proteins are required for the AQP2 redistribution in response to cAMP elevation.

AVP or forskolin stimulation of the principal cells led to a PKA-mediated increase in the phosphorylation of CDK18. This, in turn, enhances its activity [54]. The kinase activity must then be directed towards substrates other than S261 of AQP2 because the elevation of cAMP decreases the S261 phosphorylation. Such targets are currently unknown. Our CDK18 interactome analysis points to a role of protein phosphatases. We found several phosphatases associated with constitutively active CDK18, i.e., a mimic of the PKA-phosphorylated CDK18 form, and in our cell model the knockdown of CDK18 caused a reduction of phosphatase activity in response to forskolin stimulation. The control of CDK18 over phosphatase activity may be indirect through the control of phosphatase inhibitors such as Arpp19, which inhibits PP2A. [73] PP2 is associated with AQP2 [74] and involved in the control of its localization [75]. CDK18-directed phosphatases could dephosphorylate S261, explaining the decrease of the S261 phosphorylation in the presence of PKA-activated CDK18 when the cAMP level in the principal cells rises in response to AVP or as in our model through forskolin. A candidate is PP2C. Cheung et al. discovered that PP2C is required for the AVP-mediated dephosphorylation of S261 [70]. However, PP2C was not found in our CDK18 interactome analysis.

An important finding of our study is that CDK18 and STUB1 are in a complex with AQP2 and PKA. We identified STUB1 as an AKAP, directly binding PKA, and show that it mediates the interaction of AQP2 with PKA. The STUB1-associated pool of PKA was activated in response to forskolin in our principal cell model. However, this pool of PKA does not phosphorylate AQP2 or CDK18, because STUB1 knockdown did not affect PKA phosphorylation of these two proteins. STUB1 is the third AKAP tethering PKA in close proximity to AQP2. AKAP18δ and AKAP220 are the others [6,8,76]. While the knockdown of STUB1 inhibits the accumulation of AQP2 in the plasma membrane, the knockout of AKAP220 AVP-independently induces it, most likely due to its control over the small GTPase RhoA. In line with the knockdown of STUB1, the loss of AKAP220 does not alter S256 phosphorylation of AQP2 [9]. AKAP18δ interacts with PKA and the phosphodiesterase (PDE)4D, which constitutively hydrolyses cAMP and thereby terminates PKA signaling. When the AVP level rises, PKA phosphorylates and thereby enhances PDE4D activity by a factor of around two. This AKAP18δ-based signaling complex resides on AQP2-bearing vesicles and establishes a negative feedback loop maintaining low cAMP levels in the vicinity of AQP2 and terminating PKA signaling. When upon AVP stimulation the cAMP level exceeds the hydrolyzing capacity of PDE4D, AQP2 accumulates in the plasma membrane [6]. The pool/s of PKA relevant for the AVP/cAMP-induced accumulation of AQP2 in the plasma membrane must be anchored by AKAPs, as global uncoupling of PKA from AKAPs prevents the AVP-induced accumulation in the plasma membrane [10,77,78]. The STUB1-associated PKA pool is thus a prime candidate involved in controlling AQP2 trafficking. However, no pharmacological agents are available for the disruption of the interaction of PKA with a specific AKAP and thereby to determine the role of such an interaction in a defined cellular process [3,59]. In addition, it is currently unclear whether the three AKAPs, STUB1, AKAP18δ and AKAP220 reside on the same AQP2-bearing vesicles. Vesicles undergo changes in their membrane protein composition while cycling between intracellular domains and the plasma membrane, e.g., the composition of Rab proteins changes [79]. The different AKAP-PKA interactions may have different functions at different stages of vesicular trafficking. Taken together, there are distinct AKAPs and thus distinct pools of PKA in principal cells and in close vicinity of AQP2 that coordinate its localization, presumably by fulfilling different tasks.

Our study opens various new avenues. Our kinome screening provides a list of candidates whose role in the control of AQP2 is unknown. The elucidation of their function will not only reveal molecular mechanisms underlying the cAMP-triggered exocytosis-like redistribution of AQP2 to the plasma membrane. Those candidates may also be involved in other exocytic processes. The knockdown of several of the hits reduced the proliferation of our cells. Thus, such candidates have dual functions. They control localization of AQP2 and proliferation of the principal cells. In HEK293 cells, the knockdown of CDK18 caused RhoA activation [80]. Active RhoA prevents the accumulation of AQP2 in the plasma membrane of renal collecting duct principal cells [28]. Therefore, elucidating the role of CDK18 in controlling the principal cell cytoskeleton requires further studies. We provide an interactome of active CDK18 in comparison to the interactome of the kinase dead variant. A global mass spectrometry study of CMGC kinases discovered interacting proteins of wild type CDK18 [81]. Together, the data constitute a starting point for further analysis of the function of this kinase. Last but not least, our study has important clinical implications. Several disorders are characterized by a predominant localization of AQP2 in the plasma membrane. As a consequence, the patients suffer of an excessive water retention. For several of such disorders such as heart failure or ADPKD there is no satisfactory treatment available. Our data point to CDK18 as a novel pharmacological target for their treatment. Its cell type and signalosome-specific inhibition may prevent the excessive water retention in the patients.

## Figures and Tables

**Figure 1 cells-09-00673-f001:**
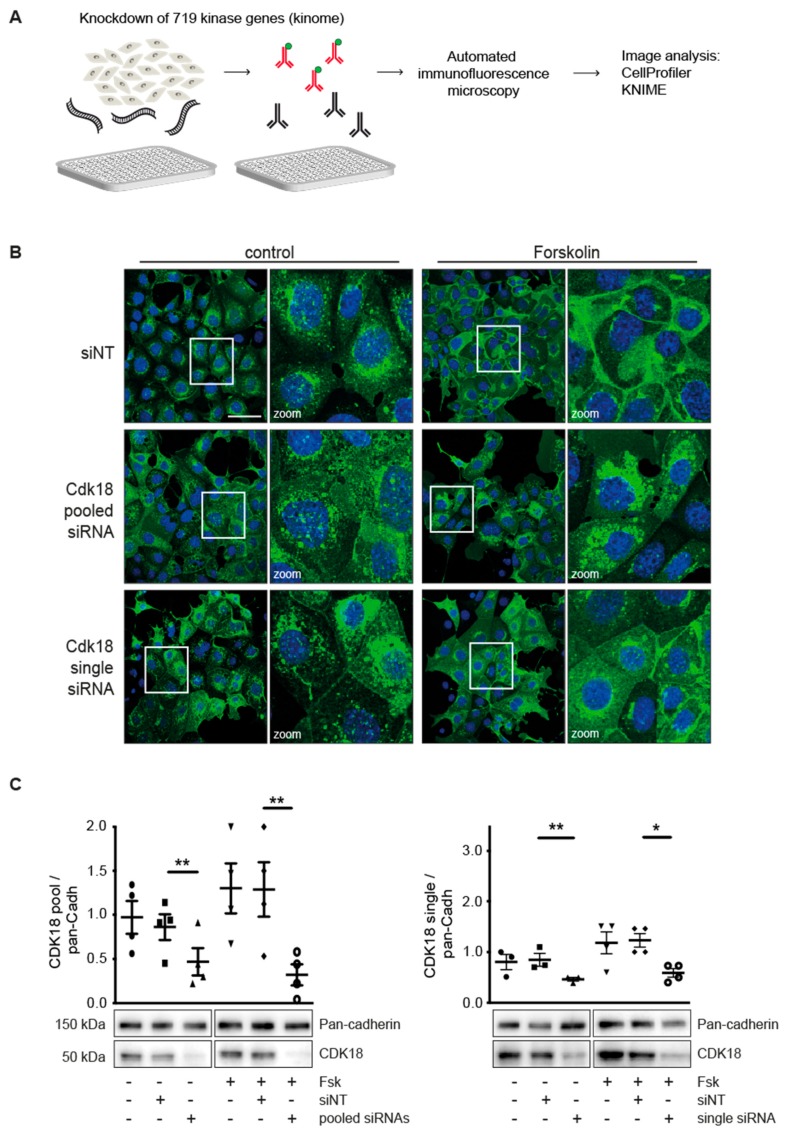
CDK18 is necessary for the cAMP-induced redistribution of AQP2 from intracellular vesicles to the plasma membrane. (**A**) Schematic representation of the Kinome-wide siRNA screening approach. MCD4 cells were seeded in 384-well microtiter plates and the expression of 719 kinases was knocked down each with a pool of four siRNAs. The effects of the knockdown on the localization of AQP2 were detected with specific anti-AQP2 and secondary Cy3-coupled antibodies and automated immunofluorescence microscopic analysis. Image analysis was carried out with CellProfiler and KNIME software. (**B**) MCD4 cells were treated with 50 nM non-targeting siRNA (siNT), a pool of four different or a single CDK18 siRNA. The cells were treated with forskolin (Fsk; 30 µM, 60 min) or were left unstimulated (control) and the localization of AQP2 was analyzed with a confocal laser scanning microscope (40× magnification). AQP2 is in green and nuclei are in blue. Shown are representative images from *n* ≥ 3 independent experiments per condition; scale bar, 50 µm. (**C**) The efficacy of the CDK18 knockdown was evaluated by Western blot analysis. CDK18 and as a loading control Pan-cadherin were detected. Signals were quantified by densitometric analysis. Statistical analysis was performed using the unpaired t-test, significant differences are indicated, * *p* ≤ 0.05, ** *p* ≤ 0.01. Mean ± SEM are plotted, *n* = 3–6 independent experiments per condition.

**Figure 2 cells-09-00673-f002:**
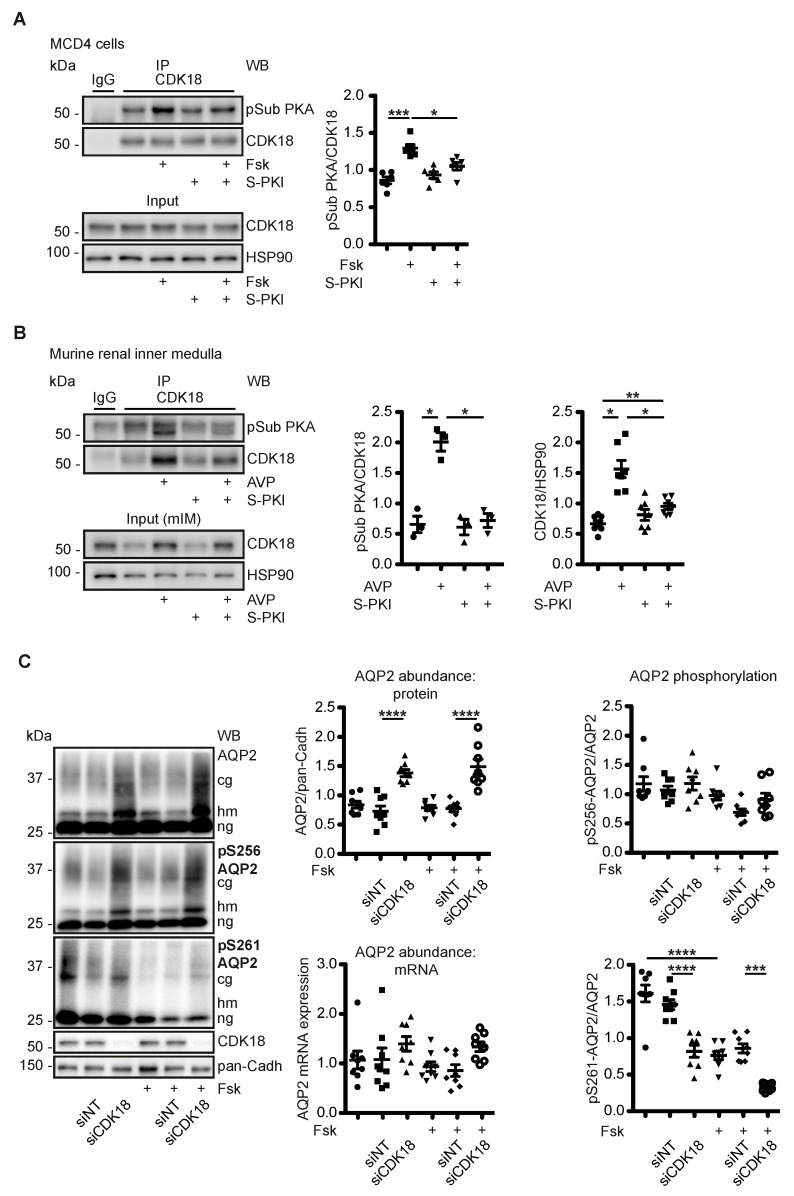
PKA modulates CDK18, which controls AQP2 phosphorylation at S261 and its abundance. (**A**) MCD4 cells or (**B**) murine inner medullae homogenates were left untreated, treated with forskolin (Fsk) or AVP and where indicated with a membrane-permeant version (stearate-coupled) of the PKA-specific heat-stable inhibitor peptide (S-PKI). CDK18 was immunoprecipitated (IP) and detected by Western blotting. The PKA-phosphorylated CDK18 (pSub CDK18) was detected with phospho-PKA substrate antibody. As a control, precipitations were carried out with unrelated IgG. As loading controls in input samples, HSP90 was detected. The Western blots were densitometrically evaluated. Shown are representative results from *n* ≥ 3 per condition. (**C**) MCD4 cells were left untransfected or transfected with non-targeting (siNT) or CDK18 siRNA, and stimulated with forskolin (Fsk) as indicated. AQP2 protein abundance and phosphorylation of its S256 and S261 were detected by Western blotting with specific antibodies. cg: complex glycosylated, hm: high-mannose, ng: non-glycosylated AQP2. Shown are representative results from *n* ≥ 8 per condition. The signals were densitometrically analysed. AQP2 mRNA expression was evaluated by PCR. Significant differences are indicated, * *p* < 0.05, ** *p* < 0.01, *** *p* < 0.001, **** *p* < 0.0001.

**Figure 3 cells-09-00673-f003:**
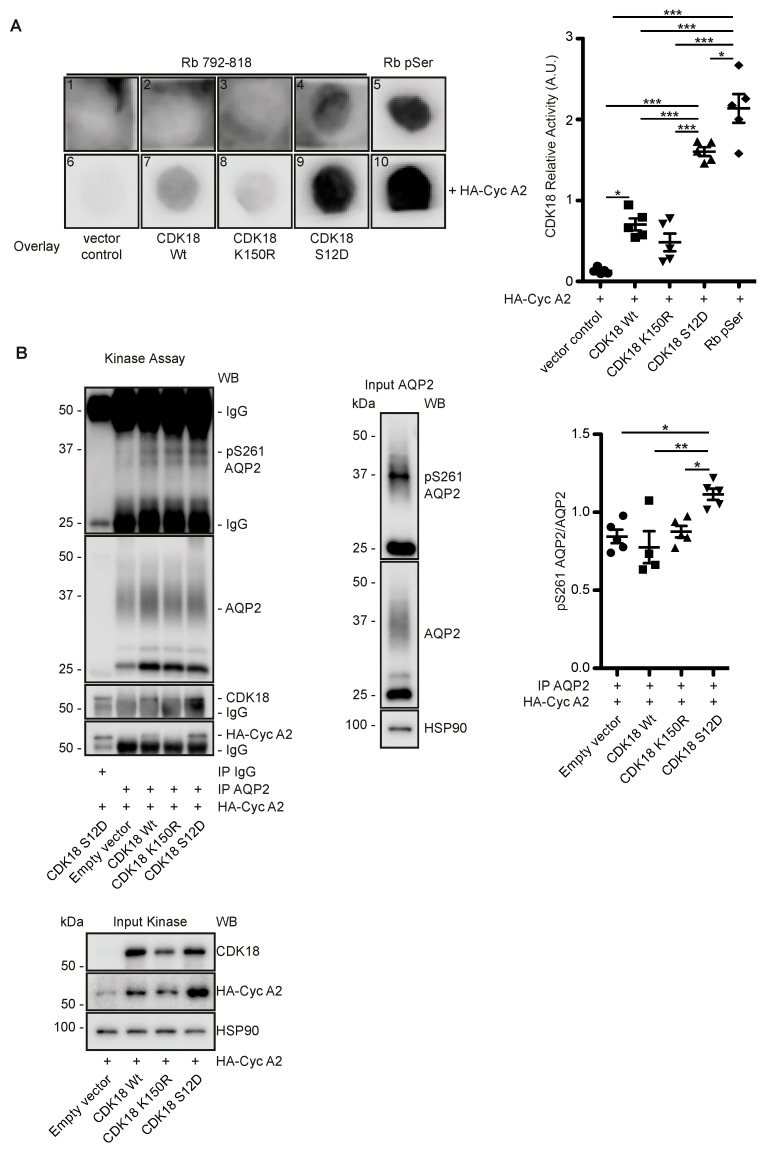
CDK18 phosphorylates AQP2 at S261. CDK18 in its wild type (Wt), kinase dead (K150R) or constitutively active form (S12D) and the co-activator Cyclin A2 (Cyc A2) were obtained by precipitation with anti-FLAG beads from HEK293 cells overexpressing the proteins. (**A**) Activity of the purified CDK18 versions was assayed using spot-synthesized peptides representing amino acids 792–818 of human retinoblastoma protein (Rb) CDK18 substrate. As a control, the peptides were spot-synthesized prephosphorylated at position S795, S807 and S811 (Rb pSer). Phosphorylated Rb was detected using an anti-phospho-CDK/MAPK substrate antibody. Due to the low signals from spots 1–5, only spots 6–10 were analyzed by densitometry; *n* = 5. (**B**) Left panel: AQP2 was immunoprecipitated (IP) from MDC4 cells. As control, immunoprecipitation was carried out with control IgG (IgG). The immunoprecipitated AQP2 was incubated with the indicated CDK18 versions obtained as in (**A**). Phosphorylated S261 of AQP2 and non-phosphorylated AQP2 was detected by Western blotting using specific antibodies (middle panel). Expression of AQP2 in MCD4 and of the CDK18 variants in HEK293 cells is shown in the input samples (lower panel). The signals of the kinase assay were semi-quantitatively analyzed by densitometry (*n* ≥ 4). Significant differences are indicated.

**Figure 4 cells-09-00673-f004:**
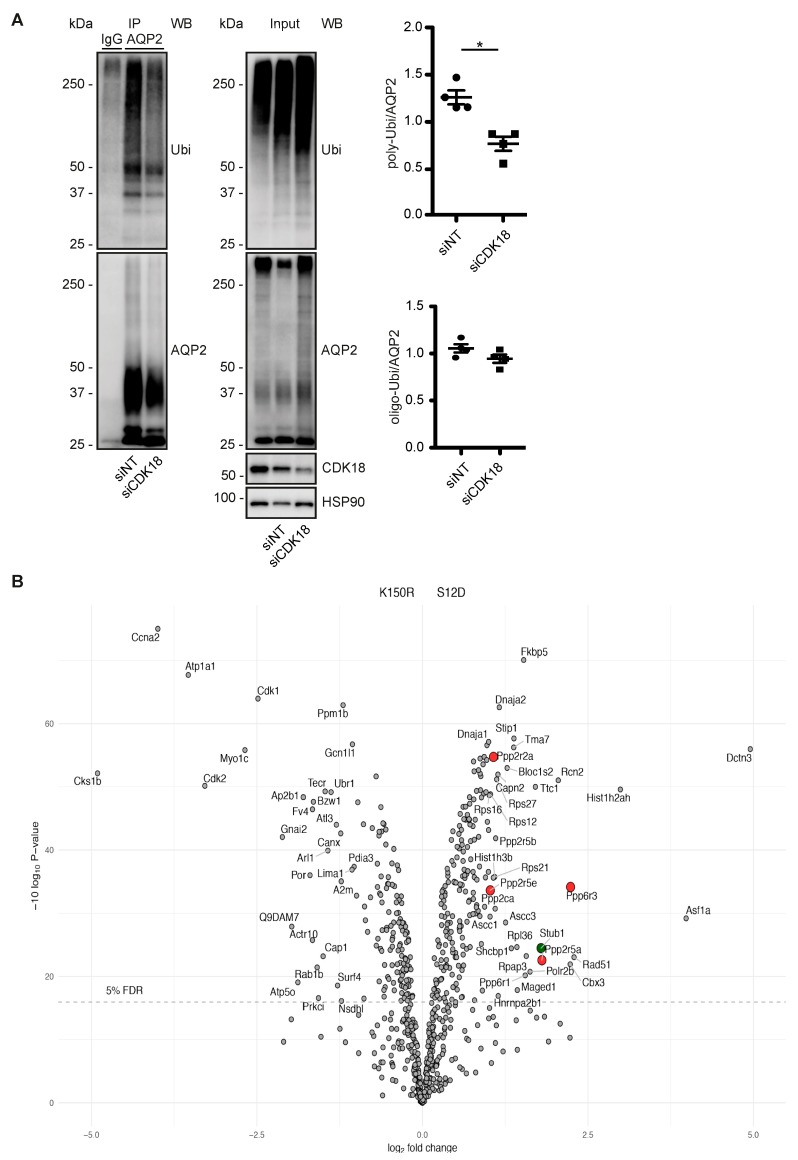
CDK18 regulates STUB1 to control AQP2 ubiquitination, abundance and localisation. (**A**) MCD4 cells were transfected with control non-targeting (siNT) or CDK18 siRNA, lysed and AQP2 was immunoprecipitated (IP). As a control, unrelated IgG was used (IgG). AQP2 and co-immunoprecipitated ubiquitin (Ubi) were detected by Western blotting (WB). Shown are a representative result from *n* = 4 independent experiments including the input samples where HSP90 was used as a loading control. The signals were semi-quantitatively analyzed by densitometry. Ubiquitin signals larger than 50 kDa were considered as poly-Ubiquitination, while under 50 kDa as oligo-Ubiquitination [25]. (**B**) CDK18-FLAG kinase dead (K150R) or constitutively active (S12D)-encoding constructs were expressed in MCD4 cells, immunoprecipitated through their Flag tags and the interactomes analysed by mass-spectrometry. The *x*-axis indicates the abundance ratios of the identified interacting proteins of K150R (negative values) versus S12D (positive values). The *y*-axis represents the significance of a two-sample moderated t-test by which the proteins were quantified. The dashed line indicates the 5% FDR cut-off based on Benjamini–Hochberg corrected *p*-values. The identified STUB1 (green) and protein phosphatases (PP, red) subunits are highlighted. (**C**) MCD4 cells were transfected with NT or STUB1 siRNA, lysed and subjected to immunoprecipitation (IP) with anti-AQP2 or control (IgG) antibodies. AQP2, CDK18, STUB1, and in the input samples HSP90 as loading control were detected by Western blotting. Shown are representative results and the densitometric semi-quantitative analysis from *n* = 4 independent experiments. Ubiquitin (Ubi) signals larger than 50 kDa were considered poly-Ubiquitination, while the ones under 50 kDa were defined as oligo-Ubiquitination [25]. (**D**) Detection of AQP2 (green) by immunofluorescence microscopic analysis of MCD4 cells transfected with siNT or STUB1 siRNA, and stimulated with forskolin (Fsk) or vehicle. Nuclei (blue) were stained with DAPI; *n* = 3; scale bar, 50 µm). (**E**) MCD4 cells were left untransfected or transfected with control siNT or CDK18 siRNA, and stimulated with forskolin (Fsk) or vehicle. Representative Western blots of the indicated proteins including the detection of cadherin (pan-Cadherin) as a loading control is shown (*n* = 4 independent experiments). The signals were semi-quantitatively analyzed by densitometry. (**F**) The STUB1 mRNA expression in the samples indicated in (**E**) was evaluated by sq-PCR (*n* ≥ 4 per condition). (**G**) MCD4 cells were transfected with either siNT or CDK18 siRNA, and treated with the proteasome inhibitor MG132 or vehicle as a control. The cells were lysed and the indicated proteins were detected by Western blotting. Shown is a representative of *n* = 5 independent experiments. The signals were semi-quantitatively analyzed by densitometry. (**H**) Mouse primary inner medullary collecting duct cells were transfected with either siNT or CDK18 siRNA. STUB1, CDK18 and HSP90 as a loading control were detected by Western blotting. Shown is the result from *n* = 5 independent experiments. The signals were semi-quantitatively analyzed by densitometry. Statistically significant differences are indicated, **p* < 0.05, ***p* < 0.01 ****p* < 0.001.

**Figure 5 cells-09-00673-f005:**
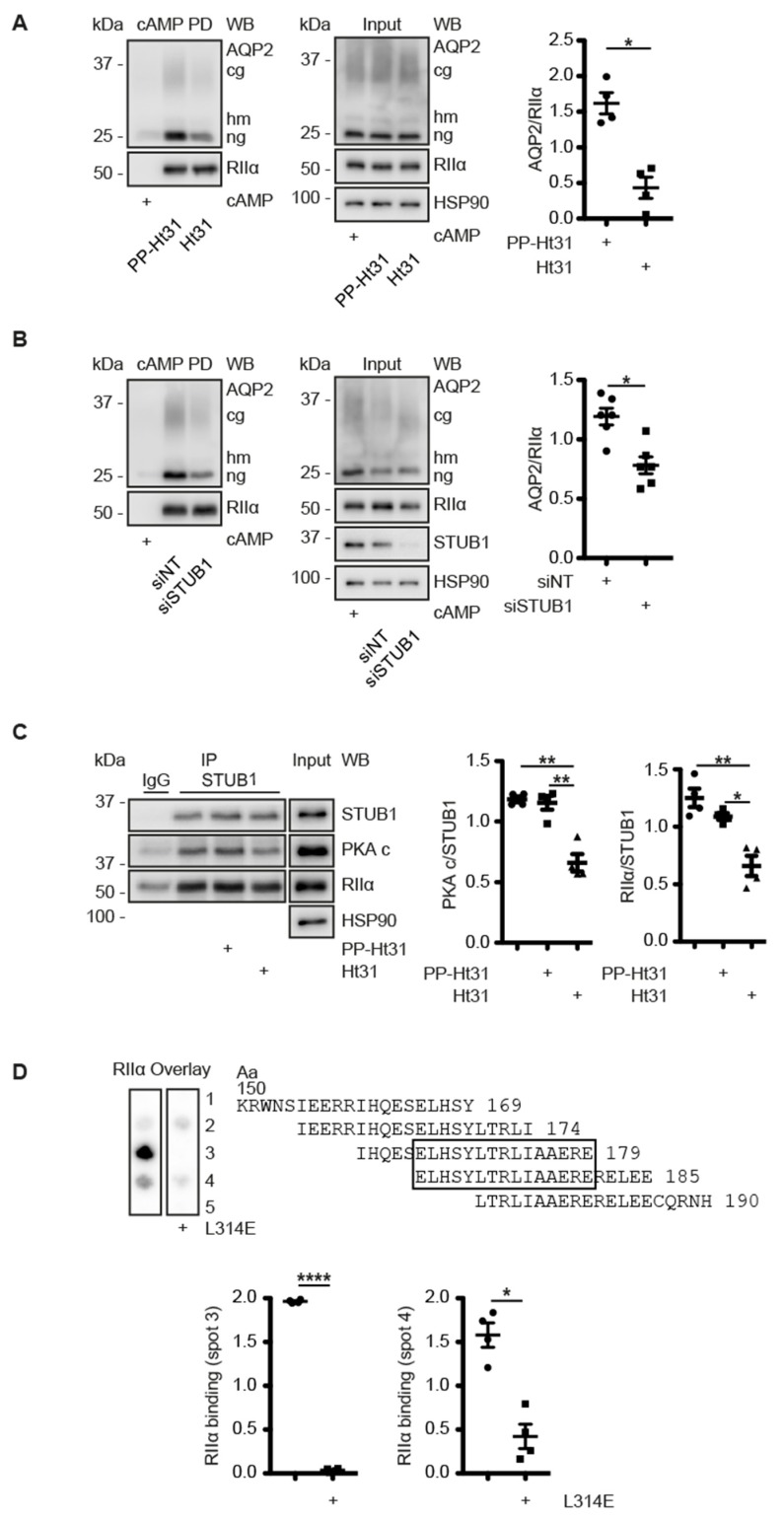
STUB1 organizes a signalosome comprising CDK18, AQP2 and PKA and functions as an A-kinase anchoring protein (AKAP). (**A**) MCD4 cells were lysed and the lysates incubated with the peptide Ht31, which inhibits AKAP-PKA interactions, the inactive peptide Ht31-PP or vehicle and subjected to cAMP-agarose pull down (PD). As negative control, the pull down samples (PD) were incubated with an excess of cAMP. The indicated proteins in the pull down and input samples were detected by Western blotting and the signals were densitometrically evaluated. Shown are representative results from *n* = 4 independent experiments. (**B**) MCD4 cells were transfected with siRNA against STUB1 or control non-targeting (siNT) and subjected to cAMP-agarose pull down. As a control, the precipitates (PD) were incubated with an excess of cAMP. The indicated proteins were detected by Western blotting and the signals densitometrically evaluated shown are representative results from *n* = 5 independent experiments. (**C**) MCD4 were lysed and the lysates were left untreated, incubated with the peptide Ht31 or PP-Ht31. STUB1 was immunoprecipitated (IP). As control, precipitations were carried out with unrelated IgG. The indicated proteins in the precipitates and input samples were analyzed by Western blotting. HSP90 served as a loading control. Shown are representative results from *n* = 4 independent experiments. The signals were semi-quantitatively analysed by densitometry. (**D**) Murine STUB1 was spot-synthesized as 20-mer peptides with 5 residues offset. Spots 1–5 cover amino acids (Aa) 150–190. The spots were incubated with P^32^-labeled recombinant human RIIα subunits of PKA in the absence or presence of the peptide L314E, which inhibits AKAP-PKA interactions. A representative overlay from *n* = 4 independent experiments is shown. The semi-quantitative densitometric analysis is shown. (**E**) MCD4 cells were left untreated or stimulated with forskolin (Fsk) and STUB1 was immunoprecipitated. As a control, unrelated IgG (IgG) was used. The indicated proteins in the precipitates and the input samples were detected by Western blotting. Shown are representative results from *n* = 3 independent experiments. The signals were semi-quantitatively analyzed by densitometry. cg: complex glycosylated, hm: high-mannose, ng: non-glycosylated AQP2. Statistically significant differences are indicated, * *p* < 0.05, ** *p* < 0.01, **** *p* < 0.0001.

**Figure 6 cells-09-00673-f006:**
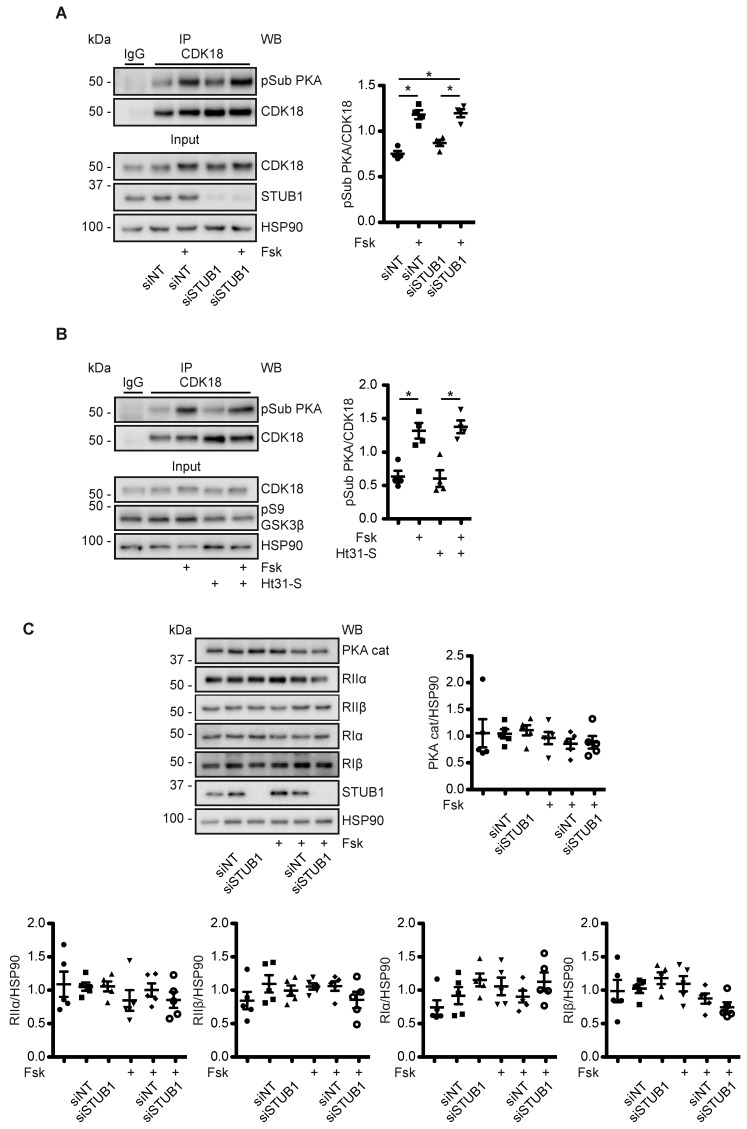
A cytosolic, STUB1-independent pool of PKA phosphorylates CDK18. (**A**) MCD4 cells were transfected with non-targeting (siNT) or STUB1 siRNA and left untreated or were treated with forskolin (Fsk) in the absence or (**B**) presence of a stearate-coupled, membrane-permeant version of the peptide Ht31, which inhibits AKAP-PKA interactions. CDK18 was immunoprecipitated (IP). As control, precipitations were carried out with unrelated IgG. The indicated proteins were detected by Western blotting. Phosphorylated CDK18 (pSub PKA) was detected with an phospho-PKA substrate antibody and CDK18 with specific antibodies; in the input samples, CDK18, STUB1, as a control for the effects of AKAP-PKA disruption, GSK3β phosphorylated by PKA (pS9), and as a loading control HSP90. Shown are representative results from *n* = 4 independent experiments and the semi-quantitative densitometric analysis. (**C**) MCD4 cells were left untreated or transfected with control non-targeting (siNT) or STUB1 siRNA, and stimulated with forskolin (Fsk) or vehicle. The cells were lysed and the indicated proteins detected by Western blot (WB) analysis. The signals were densitometrically evaluated. Shown are representative results from *n* = 5 independent experiments. Statistically significant differences are indicated, **p* < 0.05.

**Figure 7 cells-09-00673-f007:**
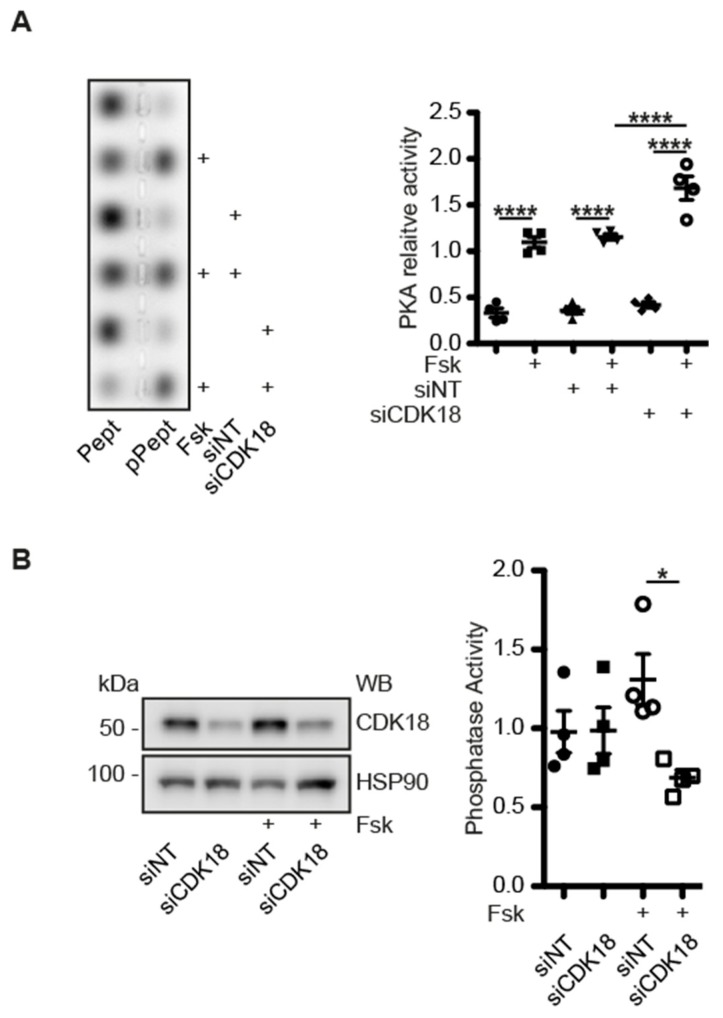
(**A**) MCD4 cells were lysed and PKA activity was determined by measuring its ability to phosphorylate a substrate peptide (PepTag A1). Shown is a representative agarose gels from *n* = 4 independent experiments with PKA-phosphorylated (pPept) and non-phosphorylated (Pept) PepTag A1 peptide. The amounts of each peptide were semi-quantitatively analysed and relative PKA activity was expressed as the ratio of phosphorylated to non-phosphorylated peptides. (**B**) MCD4 cells were transfected with siNT or CDK18 siRNA, and stimulated with forskolin (Fsk). Phosphatase activity was assayed using para-nitrophenyl-phosphate (pNPP; *n* = 4). Statistically significant differences are indicated, * *p* < 0.05, **** *p* < 0.0001.

**Figure 8 cells-09-00673-f008:**
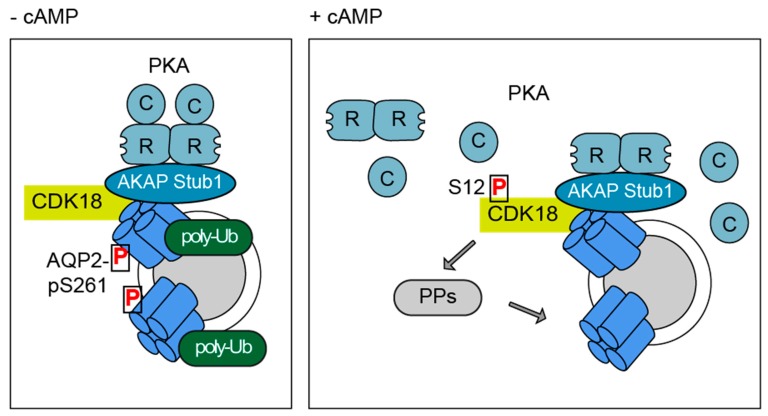
Model of the regulation of AQP2 by a signalosome comprising CDK18, STUB1 (E3 ubiquitin ligase and AKAP), PKA and AQP2. In resting renal principal cells (-cAMP), CDK18 maintains AQP2 phosphorylated at S261 and STUB1 maintains it poly-ubiquitinated. In this state, AQP2 is retained in cytoplasmic storage vesicles. Upon cAMP elevation, the complex directs a decrease of the S261 phosphorylation and poly-ubiquitination and hence facilitates the trafficking of AQP2-bearing vesicles to the plasma membrane. This process can involve dephosphorylation by protein phosphatases (PPs). The plasma membrane insertion of AQP2 promotes water reabsorption from primary urine and fine-tunes body water homeostasis.

**Table 1 cells-09-00673-t001:** Genes whose silencing prevents the cAMP-induced trafficking of AQP2 in MCD4 cells. Hits are listed according to the effects on MCD4 cell viability. Viability of non-targeting siRNA-transfected cells was considered 100%. The candidates 14–19 whose down regulation decreased MCD4 cell numbers to less than 60% were excluded from further analysis. Shown is the mean of viability from *n* = 3 independent experiments. Accession numbers refer to National Center for Biotechnology Information (NCBI, www.ncbi.nlm.nih.gov/) entries.

	Viability (%)	Protein Symbol	Gene Name	Accession
1	97	Pkia	protein kinase inhibitor alpha	NM_008862.3
2	87	Stk11	serine/threonine kinase 11	NM_011492.3
3	79	Fgfr1	fibroblast growth factor receptor 1	NM_010206.2
4	78	Ephb3	Eph receptor B3	NM_010143.1
5	75	Bmpr1b	bone morphogenetic protein receptor type 1B	NM_007560.3
6	70	Vrk3	vaccinia related kinase 3	NM_133945.1
7	70	Trim27	tripartite motif-containing 27	NM_009054.3
8	70	Cdk18	cyclin-dependent kinase 18	NM_008795.2
9	69	Ak3	adenylate kinase 3	NM_021299.1
10	69	Syk	spleen tyrosine kinase	NM_011518.2
11	66	Tk2	thymidine kinase 2 mitochondrial	NM_021028.3
12	65	Dusp2	dual specificity phosphatase 2	NM_010090.2
13	64	PFKP	Phosphofructokinase 1	NM_019703
14	52	Trpm6	transient receptor potential cation channel subfamily M member 6	NM_153417.1
15	52	Aurka	aurora kinase A	NM_011497.3
16	50	Evi5l	ecotropic viral integration site 5 like	NM_001039578.3
17	33	Aurkb	aurora kinase B	NM_011496.1
18	25	Copb2	coatomer protein complex subunit beta 2 (beta prime)	NM_015827.2
19	18	Wee1	WEE 1 homolog 1 (S. pombe)	NM_009516.3

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
