# Peer review of "Cyclin-Dependent Kinase 18 Controls Trafficking of Aquaporin-2 and Its Abundance through Ubiquitin Ligase STUB1, Which Functions as an AKAP"

_cells, 2020, doi:10.3390/cells9030673_

Round 1

Reviewer 1 Report

This is an interesting manuscript describing the identification of a signaling complex comprising AQP2/PKA/CDK18 and STUB1. As usual from this group, the work is well done and carefully analyzed. The data they have generated provide a significant and novel contribution to the field by adding one more piece to the puzzle related to the regulation of AQP2 trafficking in kidney epithelial cells. The identification of a “new” AKAP certainly adds to the apparent complexity of the process, but the authors clearly show its effect on AQP2 trafficking and its role in AQP2 protein regulation.

I have the following comments that are made to improve what is already an interesting manuscript.

  1. Throughout the text, the authors continue to make the old assumption that the main outcome of FK (and by analogy AVP) action is to cause the translocation of perinuclear AQP2 containing vesicles to the plasma membrane. Over the past few years it has become increasing evident that one of the main effects of AQP2 phosphorylation by PKA (and perhaps other kinases) is to inhibit endocytosis. The authors do mention this at one point on line 70 (where they should also include references to the original manuscripts describing this process – see below), but they also say on line 61 that  “…..and activation of PKA… are required for triggering the trafficking of AQP2 bearing vesicles to the plasma membrane”. Since it is clear that AQP2 exocytosis and membrane insertion can occur in the complete absence of any of the major phosphorylation sites in the AQP2 C-terminus (Arthur, J., Huang, J., Nomura, N., Jin, W. W., Li, W., Cheng, X., Brown, D. and Lu, H. A. J. 2015. Characterization of the putative phosphorylation sites of the AQP2 C-terminus and their role in AQP2 trafficking in LLC-PK1 cells. Am. J. Physiol. 309: F673-F679), this phrase and the thinking behind it need to be modified here and throughout the text (line 55, line 592). All that most prior reports demonstrate is that under conditions a, b or c, the amount of AQP2 on the cell surface increases – without revealing the underlying mechanism responsible for this increase. While it is true that AVP treatment can increase vesicle exocytosis, this is not dependent on S256 phosphorylation.

Lu, H. J., Sun, T. X., Bouley, R., Blackburn, K., McLaughlin, M. and Brown, D. 2004. Inhibition of endocytosis causes phosphorylation (S256)-independent plasma membrane accumulation of AQP-2.  Am. J. Physiol. Renal Physiol. 286: F233-F243.CID: PMC5582913).

Lu, H. A. J., Sun, T-X., Yi X-H., Bouley, R., McLaughlin, M., Van Hoek, A. and Brown, D. 2007. Heat-shock protein 70 interacts with aquaporin-2 and regulates its trafficking. J. Biol. Chem., 282: 28721-28732.

  1. It would be interesting to know where the STUB1 is located in relation to other “pools” of AKAPs previously identified by these authors. Did they try immunolocalization using the STUB1 antibodies. I assume they did, but they did not work. Perhaps a little more discussion on the relative roles of different AKAPs is warranted. Since elimination of STUB1 prevents AQP2 membrane accumulation, what is the residual role of AKAP18d?
  2. The pattern of AQP2 membrane accumulation appears basolateral in the images shown – it should be explicitly stated that is these cell lines, AQP2 accumulated mainly on the basolateral plasma membrane.
  3. Based on their discussion on line 571, the authors seem to be unaware of the article by Cheung et al identifying PP2A as the phosphatase that dephosphorylates S261 in response to VP stimulation. This needs to be acknowledged of course.

Cheung, P. W., Ueberdiek, L., Day, J., Bouley, R., and Brown, D. Protein phosphatase 2C (PP2C) is responsible for VP-induced dephosphorylation of AQP2 serine 261. 2017. Am. J. Physiol. Renal Physiol. 313: F404-F413.

Author Response

Reviewer 1

First of all, we would like to thank the reviewer for the kind introductory words and would like to apologise for not having included all citations.

Reviewer

  1. Throughout the text, the authors continue to make the old assumption that the main outcome of FK (and by analogy AVP) action is to cause the translocation of perinuclear AQP2 containing vesicles to the plasma membrane. Over the past few years it has become increasing evident that one of the main effects of AQP2 phosphorylation by PKA (and perhaps other kinases) is to inhibit endocytosis. The authors do mention this at one point on line 70 (where they should also include references to the original manuscripts describing this process – see below), but they also say on line 61 that  “…..and activation of PKA… are required for triggeringthe trafficking of AQP2 bearing vesicles to the plasma membrane”. Since it is clear that AQP2 exocytosis and membrane insertion can occur in the complete absence of any of the major phosphorylation sites in the AQP2 C-terminus (Arthur, J., Huang, J., Nomura, N., Jin, W. W., Li, W., Cheng, X., Brown, D. and Lu, H. A. J. 2015. Characterization of the putative phosphorylation sites of the AQP2 C-terminus and their role in AQP2 trafficking in LLC-PK1 cells. Am. J. Physiol. 309: F673-F679),this phrase and the thinking behind it need to be modified here and throughout the text (line 55, line 592). All that most prior reports demonstrate is that under conditions a, b or c, the amount of AQP2 on the cell surface increases – without revealing the underlying mechanism responsible for this increase. While it is true that AVP treatment can increase vesicle exocytosis, this is not dependent on S256 phosphorylation.

Response

We apologise for being so unclear in several sections of the manuscript. We have revised the text according to the reviewer’s suggestion and actually avoid a discussion about exo- or endocytosis because strictly speaking we have not studied these processes. The citations this reviewer suggested are included in the manuscript. We apologise for not having included them initially.

Reviewer

  1. It would be interesting to know where the STUB1 is located in relation to other “pools” of AKAPs previously identified by these authors. Did they try immunolocalization using the STUB1 antibodies. I assume they did, but they did not work. Perhaps a little more discussion on the relative roles of different AKAPs is warranted. Since elimination of STUB1 prevents AQP2 membrane accumulation, what is the residual role of AKAP18d?

Response

Indeed, the available antibodies did not work for immunofluorescence microscopy. With regard to the other AKAPs known to be expressed in renal principal cells, including AKAP18δ, we suggest in the discussion section that they are also required. They control a pool of PKA different from the one that is directed by STUB1. The different AKAPs may even reside on different pools of AQP2-bearing vesicles. Unfortunately, at present we cannot differentiate between the functions of the different AKAP-PKA interactions.

Reviewer

  1. The pattern of AQP2 membrane accumulation appears basolateral in the images shown – it should be explicitly stated that is these cell lines, AQP2 accumulated mainly on the basolateral plasma membrane.

Response

In paragraph 1 of the Results section, we have included that cAMP triggers a predominant redistribution of AQP2 into the basolateralplasma membrane. We observe that AQP2 also redistributes to the apical membrane in MCD4.

Reviewer

  1. Based on their discussion on line 571, the authors seem to be unaware of the article by Cheung et al identifying PP2A as the phosphatase that dephosphorylates S261 in response to VP stimulation. This needs to be acknowledged of course.

Response

We have included into our discussion that PP2C might be a candidate phosphatase downstream of CDK18 and included the citation, Cheung et al.

Reviewer 2 Report

The authors carried out a siRNA screen targeting 719 kinase-related genes, representing the majority of the kinases of the human genome and analyzed the effect of the knockdown on AQP2 by high-content imaging and biochemical approaches. This study is novel and interesting, but there are several comments which need to be addressed.  

1. Page 6, Line 251: The authors selected CDK18 among 13 candidates. Authors need to add a rationale for why they selected CDK18 for further studies. The cell viability after CDK18 knockdown was only 70%, which was lower than that of Pkia and Stk11 in Table 1. Thus, what was the reason why the authors did not select Pkia or Stk11 instead of CDK18?

2. Page 6, Line 251: PCTAIRE kinase 3/cyclin-dependent kinase 18 (PCTK3/CDK18) is a serine-threonine protein kinase that belongs to the CDK family. Previous findings suggested that PCTK3 may also be involved in the regulation of actin cytoskeleton dynamics by controlling the activities of Rho GTPases. Since authors demonstrated that knockdown of CDK18 inhibits the cAMP-induced redistribution of AQP2, this issue regarding the regulation of actin cytoskeleton dynamics could be added in the Discussion section, in addition to the findings of AQP2 phosphorylation.

3. Page 9, Line 302 and Figure 2C: siRNA-mediated knockdown of CDK18 was associated with a decreased S261 phosphorylation of AQP2, even in the absence of forskolin stimulation. The decrease of S261 phosphorylation was more prominent when MCD4 cells with siRNA-mediated knockdown of CDK18 were treated by forskolin. Then, what about the changes in MAPK activity or their phosphorylation status (e.g. p38, ERK, and JNK) in these conditions?

4. Figure 2C: Forskolin treatment did not induce AQP2 phosphorylation at Serine 256 in untransfected MCD4 cells and siNT-transfected MCD4 cells. Since forskolin treatment (30 uM, 60 min) could be potent to induce AQP2 phosphorylation at Serine 256, this finding should be explained.

5. Page 12, Line 350: To identify the E3 ligase that is regulated by CDK18, authors compared the interactome of the kinase dead and constitutively active form of CDK18 forms by mass spectrometry. Then the authors picked up E3 ligase STUB1 (Figure 4B). Did authors find other E3 ligases, such as NEDD4, NEDD4L, AMFR, etc, as previously demonstrated by other studies about E3 ligases and AQP2 degradation (Medvar et al, PMID: 27199454; Lee et al, PMID: 21734099; Trimpert et al, PMID: 28931009)?

6. Figure 4D: The effects of siSTUB1 was unclear. The labeling intensity as well as targeting to the plasma membrane of AQP2 were weaker than that of control siNT-treated MCD cells and most of the labeling was seen in the perinuclear region. The AQP2 labeling in Figure 4D should be confirmed.  

Author Response

Reviewer 2

  1. Page 6, Line 251: The authors selected CDK18 among 13 candidates. Authors need to add a rationale for why they selected CDK18 for further studies. The cell viability after CDK18 knockdown was only 70%, which was lower than that of Pkia and Stk11 in Table 1. Thus, what was the reason why the authors did not select Pkia or Stk11 instead of CDK18

Response

The main reason for the decision to characterise the role of CDK18 was to elucidate a new signalling pathway controlling AQP2. We would have expected that characterisation of PKIA or Stk11 would only extend existing knowledge on pathways regulating AQP2. PKIA blocks PKA activity by binding to its catalytic subunits upon their cAMP-dependent dissociation from its regulatory subunits1. It regulates the nuclear export of the free catalytical PKA subunits2. STK11 controls the activity of AMP-activated protein kinases (AMPK) and thus, is involved in various processes such as cell growth, energy metabolism and cell polarity3. An involvement of PKA and AMPA in AQP2 regulation was already known. We have included this reasoning into the results section.

Reviewer

  1. Page 6, Line 251: PCTAIRE kinase 3/cyclin-dependent kinase 18 (PCTK3/CDK18) is a serine-threonine protein kinase that belongs to the CDK family. Previous findings suggested that PCTK3 may also be involved in the regulation of actin cytoskeleton dynamics by controlling the activities of Rho GTPases. Since authors demonstrated that knockdown of CDK18 inhibits the cAMP-induced redistribution of AQP2, this issue regarding the regulation of actin cytoskeleton dynamics could be added in the Discussion section, in addition to the findings of AQP2 phosphorylation.

Response

We have addressed this point in the last paragraph of the discussion where we direct the reader to observations by Matsuda et al., who observed that the knockdown of CDK18 causes activation of Rho.4

Reviewer

  1. Page 9, Line 302 and Figure 2C: siRNA-mediated knockdown of CDK18 was associated with a decreased S261 phosphorylation of AQP2, even in the absence of forskolin stimulation. The decrease of S261 phosphorylation was more prominent when MCD4 cells with siRNA-mediated knockdown of CDK18 were treated by forskolin. Then, what about the changes in MAPK activity or their phosphorylation status (e.g. p38, ERK, and JNK) in these conditions?

Response

In the second paragraph of the Discussion section, lines 593-596, we point out that the incomplete reduction of the S261 phosphorylation by CDK18 knockdown may be the result of phosphorylation through other kinases that have been shown to also phosphorylate this site.

Reviewer

  1. Figure 2C: Forskolin treatment did not induce AQP2 phosphorylation at Serine 256 in untransfected MCD4 cells and siNT-transfected MCD4 cells. Since forskolin treatment (30 uM, 60 min) could be potent to induce AQP2 phosphorylation at Serine 256, this finding should be explained.

Response

This observation is in line with the finding that the plasma membrane accumulation of AQP2 can occur even in the absence of PKA.5We have included this interpretation in the Results section, lines 337-340.

Reviewer

  1. Page 12, Line 350: To identify the E3 ligase that is regulated by CDK18, authors compared the interactome of the kinase dead and constitutively active form of CDK18 forms by mass spectrometry. Then the authors picked up E3 ligase STUB1 (Figure 4B). Did authors find other E3 ligases, such as NEDD4, NEDD4L, AMFR, etc, as previously demonstrated by other studies about E3 ligases and AQP2 degradation (Medvar et al, PMID: 27199454; Lee et al, PMID: 21734099; Trimpert et al, PMID: 28931009)?

Response

We have not found association of CDK18 with other E3 ligases. Other E3 ligases that are involved in controlling AQP2 may reside in other protein complexes. Also, deubiquitination enzymes such as deubiquitylases USP4 were not found.

Reviewer

  1. Figure 4D: The effects of siSTUB1 was unclear. The labeling intensity as well as targeting to the plasma membrane of AQP2 were weaker than that of control siNT-treated MCD cells and most of the labeling was seen in the perinuclear region. The AQP2 labeling in Figure 4D should be confirmed.  

Response

The increase of the AQP2 signal in the perinuclear region is what we repeatedly observed and what resembled the result of the CDK18 knockdown (Fig. 1). We thank the reviewer for pointing out the low intensity in the siSTUB1-treated cells. We have replaced Fig. 4D with other representative images.

Round 2

Reviewer 2 Report

I have read the authors' reply and revised manuscript. I would like to accept the revised manuscript and recommend this manuscript to be published.